# Payload-Based Traffic Classification Using Multi-Layer LSTM in Software Defined Networks [†]

**Hyun-Kyo Lim [1]** , **Ju-Bong Kim [2]**, **Kwihoon Kim [3]**, **Yong-Geun Hong [3]** and **Youn-Hee Han [2],***

[1] Interdisciplinary Program in Creative Engineering, Korea University of Technology and Education, Cheonan 31253, Korea; glenn89@koreatech.ac.kr

[2] Department of Computer Science & Engineering, Korea University of Technology and Education, Cheonan 31253, Korea; rlawnqhd@koreatech.ac.kr

[3] Electronics and Telecommunications Research Institute, Daejeon 34129, Korea; kwihooi@etri.re.kr (K.K.); yghong@etri.re.kr (Y.-G.H.)

**\*** Correspondence: yhhan@koreatech.ac.kr; Tel.: +82-10-3912-0900

[†] This paper is an extended version of the conference paper presented in the 1st International Conference on Artificial Intelligence in Information and Communication (ICAIIC 2019) [40].

**Abstract:** Recently, with the advent of various Internet of Things (IoT) applications, a massive amount of network traffic is being generated. A network operator must provide different quality of service, according to the service provided by each application. Toward this end, many studies have investigated how to classify various types of application network traffic accurately. Especially, since many applications use temporary or dynamic IP or Port numbers in the IoT environment, only payload-based network traffic classification technology is more suitable than the classification using the packet header information as well as payload. Furthermore, to automatically respond to various applications, it is necessary to classify traffic using deep learning without the network operator intervention. In this study, we propose a traffic classification scheme using a deep learning model in software defined networks. We generate flow-based payload datasets through our own network traffic pre-processing, and train two deep learning models: 1) the multi-layer long short-term memory (LSTM) model and 2) the combination of convolutional neural network and single-layer LSTM models, to perform network traffic classification. We also execute a model tuning procedure to find the optimal hyper-parameters of the two deep learning models. Lastly, we analyze the network traffic classification performance on the basis of the F1-score for the two deep learning models, and show the superiority of the multi-layer LSTM model for network packet classification.

**Keywords:** traffic classification; recurrent neural network; long short-term memory; convolutional neural network; software defined networks

## 1. Introduction

Recently, the importance of network operation and management has been emphasized due to the emergence of various services and applications. In particular, due to the rapid growth of the Internet of Things (IoT), various related applications and services are being provided through the network. Therefore, network packets or flows should be differentiated according to applications or services provided through the network. In particular, video and voice services require fast transmission. On the other hand, text services can provide adequate performance without fast transmission. In addition, peer-to-peer (P2P) services, such as BitTorrent, account for a significant proportion of the global Internet traffic and, thus, have a significant impact on the overall network speed. Therefore, the IoT network operators try to provide smooth quality of service (QoS) by assigning different priorities according to each service.

Network traffic classification [1–4] for providing different QoS according to each application has been being actively researched. There are various methods for network traffic classification, such as rule-based, correlation-based, and payload-based methods. Rule-based methods are widely used for network traffic classification [5–7]. They classify packets entering the network according to predefined rules. The classification methods usually use the header information of the network packet, and they are performed on the basis of the source and destination IP addresses and port numbers of the packet headers. Although these methods classify network traffic by well-known application port numbers or specific rules based on packet header information, the traffic is not well classified for unknown applications because network operators do not know the port number or specific rules in advance. Moreover, it is inconvenient for a network operator to manually add a rule or port number to provide a new network service.

Correlation-based network classification methods classify network traffic by selecting packets with high correlation between traffic packets considering the correlation between network traffic. They usually find statistical flow characteristics in network traffic flow, and incorporate them into machine learning techniques, e.g., logistic regression, decision tree, and support vector machine [8–10]. As such statistical characteristics, packet size, packet arrival rate, flow duration, and inherent business is usually used. It has been reported that the classification accuracy is relatively high. However, they require additional calculations for correlations in each flow, and additional consumption occurs when constructing datasets based on the correlation.

Payload-based network classification methods classify network traffic by using the pure application layer payload information excluding the packet header information of the entire network traffic [11–14]. Currently, it is no longer accurate to classify network traffic with packet header information containing the well-known port numbers or IP addresses [15], since many IoT devices and mobile devices use private or dynamic IP addresses and variable port numbers. Payload-based network classification methods overcome the IP address and port number dependency problem because they are not affected even if the header information is changed. They are usually deployed with deep packet inspection (DPI) techniques. Through DPI, it is possible to compare the contents of packet payload against a set of rules, which are usually written in a string format. However, the usage of such format rules imposes strong limitations such as limited expressiveness and inability to cope with various complex services [16].

The study in the field of machine learning have progressed actively, and machine learning has been adopted in various areas [2]. Thus, network traffic classification using machine learning has been researched extensively [17–29]. Additionally, network traffic classification methods using machine learning in software defined networks (SDNs) are actively researched [21–29]. In particular, deep learning models [30–32] have been recently studied for network traffic classification since their performance is known to be usually better than other machine learning algorithms [27–29]. In the existing network traffic classification methods based on deep learning models [27,28], however, classification is usually performed by the header information as well as the payload of packets as the learning feature. The limitation of such an approach can arise in case the head information is collected into a limited dataset collected within a local network and the deep learning model training is performed with such a dataset. In a real network that extends beyond the local limits, it is difficult to perform classification well by using the previously trained model, since the header information is usually varied in the real network due to tunneling, network address translation, security policy, etc.

In this study, we propose a payload-based traffic classification using deep learning models in SDNs for providing efficient QoS or scheduling for each application. In SDNs, the separated control plane can apply routing or QoS or scheduling decisions to the network equipment of the data plane. We add new deep learning-based classifier modules to the control plane and allow the classification results to be used for determining and applying appropriate routing or QoS or scheduling policy into data plane. We develop a traffic data preprocessing method to create deep learning datasets by using the only payload of packets, train deep learning models with the datasets, and evaluate the

performance of the models. Our goal of deep learning using only payload information is to let the learned model fit of unseen packets well. A model with strong generalization ability can fit the whole data sample space well. Excluding header information that has a typical structure and inconsistent values helps improve such generalization performance.

We treat the payload of packets as image data, which have been actively used as deep learning dataset in the artificial intelligence research area, so that we try to apply the representative deep learning models into the network traffic classification problem and investigate their performance. The imaged packets are collected for each application and flow-based datasets are constructed from them. The two deep learning models, which include the multi-layer long short-term memory (LSTM) model and the combination of convolutional neural network (CNN) and single-layer LSTM models (hereinafter called CNN + LSTM) [28], are trained to classify network traffic using the generated datasets. The multi-layer LSTM and CNN + LSTM models are suitable for learning sequential datasets, and the two models are trained with the flow-based datasets that include the sequential data of network traffic. We also execute a model tuning procedure to find the optimal hyper-parameters of each deep learning model and enhance the performance of network traffic classification. Then, we compare the performance of the two deep learning models on the basis of the F1-score measure, and demonstrate the effectiveness of the models.

The remainder of this paper is organized as follows. In Section 2, we review related studies and state the motivation for our work. In Section 3, we propose an SDN-based network architecture with deep learning sub-system for traffic classification. In Section 4, we explain the data preprocessing step. In Section 5, we describe the two deep learning model architectures for network traffic classification. In Section 6, we introduce the model tuning method for searching the optimal hyper-parameters. In Section 7, we present and analyze the experimental results on the performance of the two models. In Section 8, we describe how to classify traffic flow using the learned model in the flow classifier of the proposed SDN-based network architecture. Lastly, we provide concluding remarks in Section 9.

## 2. Related Work

Many studies have focused on network traffic classification technologies. Classical studies involve rule-based or statistical correlation-based network traffic classification. Machine learning has been researched extensively and studies on network traffic classification using machine learning have been actively conducted [18,19,21–29].

Shafiq et al. [17] attempted to classify network traffic by machine learning using different kinds of datasets. They used the three ML algorithms: multi-layer perceptron, C4.5 decision tree, and support vector machine. As a result, the C4.5 decision tree algorithm showed better performance than the other two algorithms. Singh et al. [18] used an unsupervised machine learning approach for network traffic classification. In this paper, the unsupervised K-means and the expectation maximization algorithm were used to cluster the network traffic application based on the similarity between them.

SDN is an emerging networking paradigm that gives hope to change the limitations of current network infrastructures. SDN as a concept separates data plane and control plane to confront limitations and challenges of today's networking. In particular, since the SDN is logically centralized, controllers have a global visibility of the whole network unlike current networking. Furthermore, traffic scheduling and QoS control become easier and feasible for network administrators. Huang et al. [19] considered traffic scheduling in SDN where several DPI proxy nodes are available for serving flows from ingress switches. Additionally, they studied a problem to minimize the delay of DPI processing, and designed a two-phase algorithm that can quickly select proxy and find routing paths for incoming flows. They also proposed a rule multiplexing scheme [20], in which a set of rules deployed on each node apply to the whole flow of a session going through yet toward different paths to deal with an efficient rule placement in QoS guaranteed multipath routing.

Additionally, research on traffic classification using traditional machine learning is actively underway. Parsaei et al. [21] introduced a network traffic classification that occurs in software-defined

networking (SDN). Four neural network estimators were used to classify traffic from SDN networks: Feedforward Neural Network, Multi-layer Perceptron (MLP), Levenberg-Marquardt (NARX), and Native Bayes (NARX). The four Neural Network Estimators provided 95.6%, 97%, 97%, and 97.6% in terms of accuracy, respectively. Yu et al. [22] proposed a novel SDN flow classification framework using DPI and semi-supervised learning multiple classifiers. Three types of mechanisms known as Heteroid Tri-Training, Tri-Training, and Co-Training were used as semi-supervised multiple classifiers. Based on the result of semi-supervised multiple classifier, the proposed architecture classified network traffic flows into different QoS categories. Amaral et al. [23] deployed a simple architecture in an enterprise network that gathers traffic data using the OpenFlow protocol. After the traffic collected through the OpenFlow protocol was generated as training data, the network traffic was classified using Random Forests, Stochastic Gradient Boosting, and Extreme Gradient Boosting algorithms. However, these previous works do not utilize recent deep learning models. Most traditional machine learning requires a feature engineering process to reduce the complexity of data and find patterns in the dataset. The feature engineering requires experts to create a process for identifying data and extracting patterns themselves, and takes a long time. However, the deep learning models reduce the task of extracting new features from all input data. In general, the deep learning models learn the low-level features of input data in the initial layer and then the high-level representation of data in the layer. In addition, deep learning models perform far better than traditional machine learning algorithms as the amount of learning data increases. The deep learning method has achieved very good performance in many domains including image classification and speech recognition [11,12].

Recently, network traffic classification using deep learning has begun to be studied. Wang et al. [27] used a CNN model to classify malware traffic and general traffic. First, if a five-tuple (source IP/port, destination IP/port, protocol) is the same among the packets. One flow is defined as one dataset. The constructed dataset is used to train the CNN model to classify malware traffic and general traffic. The accuracy to classify malware traffic and general traffic was not low. Lopez-Martin et al. [28] performed network traffic classification using a deep learning model that combines CNN and LSTM. The packet data, extracted from the header information and payload data in the packet using the DPI tool, was used as the learning data. The extracted learning data was used as input data for the combined model of CNN and LSTM. With datasets including the short length of packets in a flow, they showed that the combination of CNN and LSTM models perform better than the individual CNN and LSTM models. However, these previous works include packet header information in their traffic dataset, while we use a payload-based dataset excluding packet header information in this paper. In addition, we use the multi-layer LSTM model with deeper layering architecture.

In Reference [29], we already used payload-based datasets excluding the IP and TCP/UDP headers, and developed CNN and ResNet models for traffic classification. The results of network traffic classification show that CNN and models have the F1-score values of 0.95625 and 0.96875, respectively, and ResNet model is superior to the CNN model.

In this study, we extend the previous work and develop a new multi-layer LSTM model for network traffic classification when the datasets consist solely of the payload of the packets. In addition, this study shows that the multi-layer LSTM model performs better than the CNN and LSTM model (proposed by Reference [28]). In addition, we propose a new SDN-based network architecture that can generate flow rules according to QoS of various applications by utilizing the network traffic classification result of the learned deep learning model.

## 3. Proposed SDN Architecture for Deep Learning-Based Traffic Classification

Our proposed network architecture for traffic classification is configured on an SDN architecture. The SDN separates the data plane from the control plane. The data plane remains only in the network equipment and focuses only on packet transmission, while the control plane determines the packet routing path or QoS policy in a central external controller. It can use the OpenFlow protocol for communication between the two separated planes. In addition, the use of open source APIs reduces

the dependence on network equipment companies and enables the development and use of various network software.

Figure 1 shows the proposed SDN architecture for traffic classification. In the proposed architecture, the new two sub-systems, 'labeled packet collector' and 'classifier,' are located in the data plane and the control place, respectively. On the data plane, the labeled packet collectors collect and label the network packets generated by the application program run on end-hosts. In addition, the labeled packet collector sends to the classifier the bundled flow dataset in which the packets with the same five attributes, source IP, source Port, destination IP, destination Port, and protocol, are bundled into the same flow, and a label is also attached into the flow. In the classifier, the training dataset is generated through the feature selector and data preprocessing process. The generated dataset is also used for the learning process of multi-layer LSTM or CNN + LSTM model as well as parameter tuning of the models.

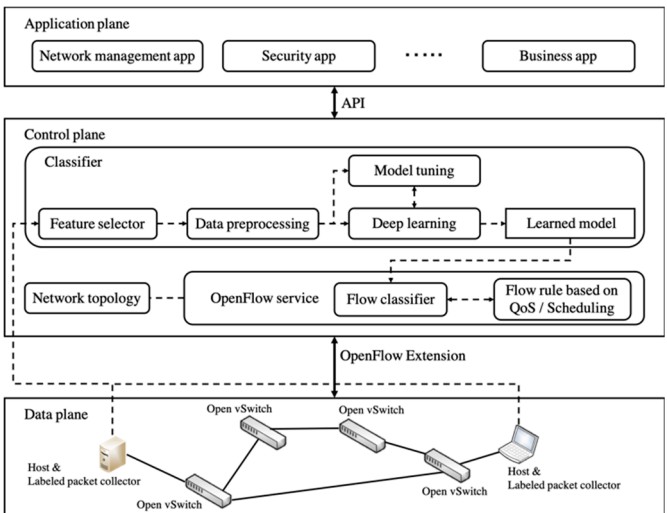

**Figure 1.** Proposed architecture in software defined network (SDN)-based Network.

The learned model is used to classify network traffic in the flow classifier. In Section 8, we describe in more detail how the flow classifier classifies network traffic using the learned model. Then, appropriate flow rules can be generated according to the QoS or scheduling policy of each application by using the classification results. Such flow rules are applied into OpenFlow switches (e.g., Open vSwitches) of data plane. Afterward, the Open Switches forward network traffic based on new flow rules. Overall, SDN-based architecture proposed in this section can classify traffic using a deep learning model and generate appropriate flow rules, according to QoS or scheduling policy of each application.

## 4. Data Preprocessing

In this study, the labeled Packet Capture (PCAP) traces provided by the UPC's Broadband Communications Research Group [33] is used for training and testing the deep learning models. The PCAP trace file capture and store network packets using programs such as Wireshark and tcpdump. The size of the original PCAP trace file provided is around 59 GB, with a total of 769,507 flows in the file. An additional information file provided with the PCAP trace file includes the labels (application name) of the traffic data and provides the ground truth for the prediction based on deep learning (To collect and accurately label the traffic flows, a volunteer-based system was used. For the details of it, refer to Reference [33]).

For the preprocessing of the supplied data, based on the number of flows, eight types of applications with more than 1000 flows were selected. The eight applications' label names are the Remote Desktop Protocol (RDP), Skype, SSH, BitTorrent, HTTP-Facebook, HTTP-Google, HTTP-Wikipedia,

and HTTP-Yahoo. The application layer payload data of the selected applications are filtered and extracted, and the learning data are generated using the extracted payload data.

*Learning Data Generation*

This subsection describes the process to convert the above-mentioned application layer payload data into the learning data suitable for deep learning models.

The overall learning data for each application are generated by arbitrarily extracting packets of eight applications (RDP, SSH, Skype, BitTorrent, Facebook, Wikipedia, Google, and Yahoo) from the application layer payload data. In each application, random flow's packets were extracted. All bits in the payload data of a packet are divided by 4 bits and grouped into one pixel of imaged data. Therefore, one pixel of the imaged data represents the decimal numbers 0 (=0x0000) to 15 (=0x1111). According to the pre-defined image size values, 36 (= 6 × 6), 64 (= 8 × 8), 256 (= 16 × 16), and 1024 (= 32 × 32), the pixels of one image data are taken from the beginning of each packet, and the size of one image data is readjusted to 36, 64, 256, and 1024 pixels. Figure 2 shows the case of the imaged packet data of 256 pixels for an arbitrary packet of each application. If the extracted payload size is smaller than the pre-defined size, the image is adjusted by zero-padding to match the pre-defined size.

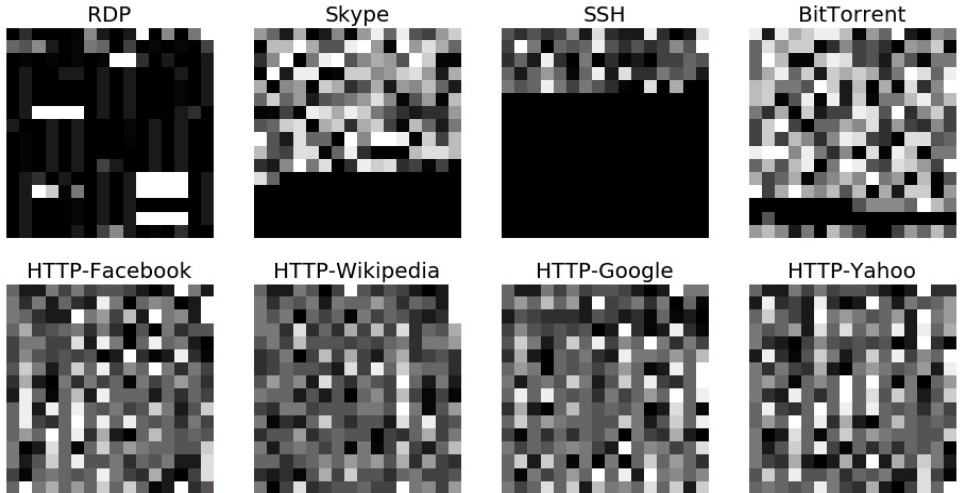

**Figure 2.** Sampled images of network packets extracted from the Packet Capture (PCAP) data.

The application layer payload data is converted into the flow-based dataset. In this study, a network flow is defined as a unidirectional sequence of packets between two endpoints with five common attributes as well as the source IP/port, destination IP/port, and protocol. The flow-based learning dataset is also arbitrarily selected for each application in the application layer payload data. One selected flow-based data consists of a series of the first N packets in the corresponding flow. The number N is set to 30, 60, and 100. Each of the packets is pre-processed in the same manner as previously described and converted into imaged packet data. For each of the eight applications, we extract 2000 flows from the application layer payload data, and the flow-based learning dataset has a total of 16,000 flows.

Lastly, for target data of eight applications, each label is expressed as a one-hot vector with eight lengths. A one-hot vector is a 1 × 8 matrix with all 0s and a single 1 used to distinguish the label representing an application.

## 5. Deep Learning Models

This section describes deep learning models used to classify network traffic. The selected deep learning models are LSTM and CNN + LSTM [28] in this study. The LSTM model is specialized for repetitive and sequential data learning. Therefore, the previous learning data is reflected in the current

learning data using the circulation structure. It is generally used for the composition of speech, wave, and text. In the CNN + LSTM model, the convolution layer extracts the features from the original input data, and then the features are used for the input data to the LSTM model. That is, artificially refined sequential data is used in the classification work in the CNN + LSTM model. In this study, the two models are used to classify flow-based learning data that contain sequential information of network traffic.

### 5.1. The Multi-Layer LSTM Architecture

An LSTM [34,35] is a network architecture that can accept the arbitrary length of inputs, and it can be implemented flexibly and in various ways, as required. Therefore, the LSTM architecture used in this study is composed of multiple layers, as shown in Figure 3. In the multi-layer LSTM model, a number of sequential packets per flow (30, 60, or 100) are received at the input layer to train the flow-based dataset. In the flow-based dataset, the first packet of one flow is input to the first cell of the LSTM layer. The result obtained from the first LSTM cell is used as input at the time of arrival of the next packet at the input. Therefore, the result of the first cell affects the operation of the second cell. Furthermore, the operation result of the first cell is also used as the input of the second LSTM layer. At the same time, the operation result of the first cell of the second LSTM layer is used as the input of the second cell of the layer. Afterward, the operation result of the second cell of the third LSTM layer is used as the input of the third cell of the layer. The final target output at the end of the sequence is a label classifying the eight applications through the output layer.

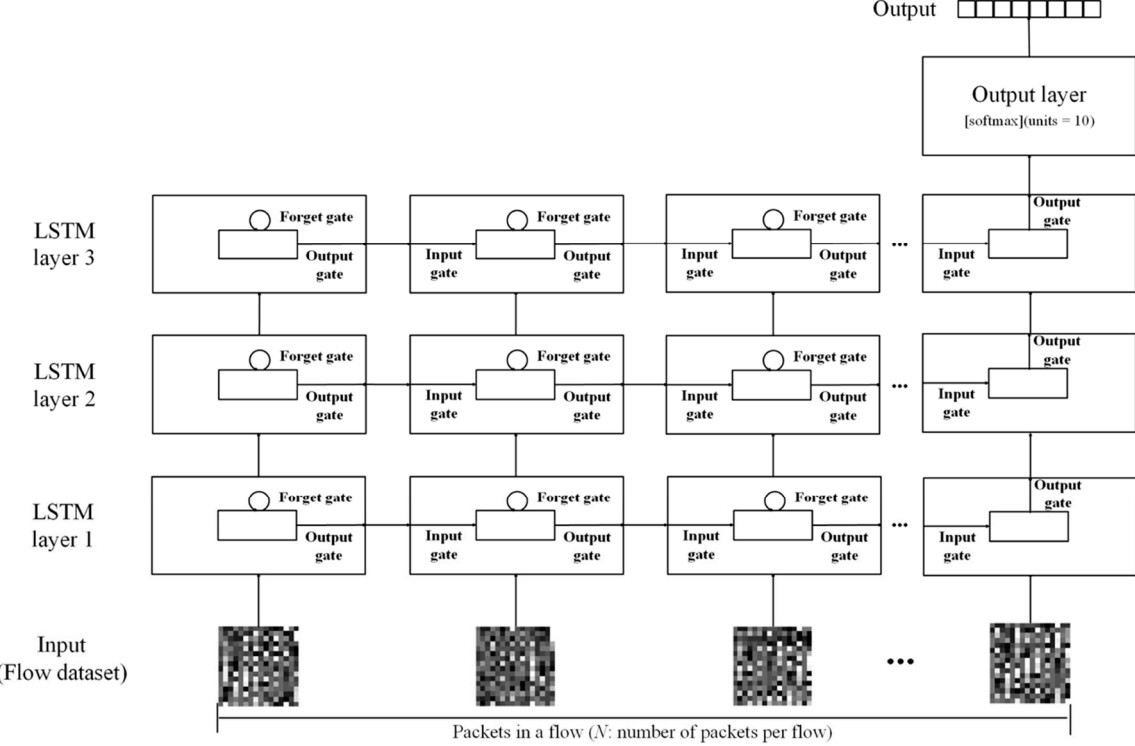

**Figure 3.** The long short-term memory (LSTM) learning model architecture. It consists of three LSTM layers.

The LSTM model also determines whether the weight value is maintained by adding another feature layer called a cell state in an LSTM cell. The LSTM model has the ability to remove or add information to the cell state, which is carefully regulated by structures called gates. Gates are a way to optionally let information through and they are responsible for adding or deleting past information,

so that LSTM is pretty persistent. The LSTM model can control the long-term memory as well as the result, so that it works a lot better for most tasks.

### 5.2. The CNN and LSTM Combination Network Model Architecture

The learning model combining CNN and LSTM is a combination of two convolution layers of the CNN model and one LSTM layer (This architecture is the same as the one of the best model provided by Reference [28]). As shown in Figure 4, when the flow-based dataset is input to the convolution layer, the image data are compressed through the filters of the layer. The result of the convolution layer is adjusted to the input size of the LSTM layer through a reshaping process before it enters the input of the LSTM layer. As in the previous LSTM learning process, the result of the first cell affects how the second input is calculated. When the final input data comes in, the calculation of the last cell is affected by the result of the previous cells, and the final classification result is extracted.

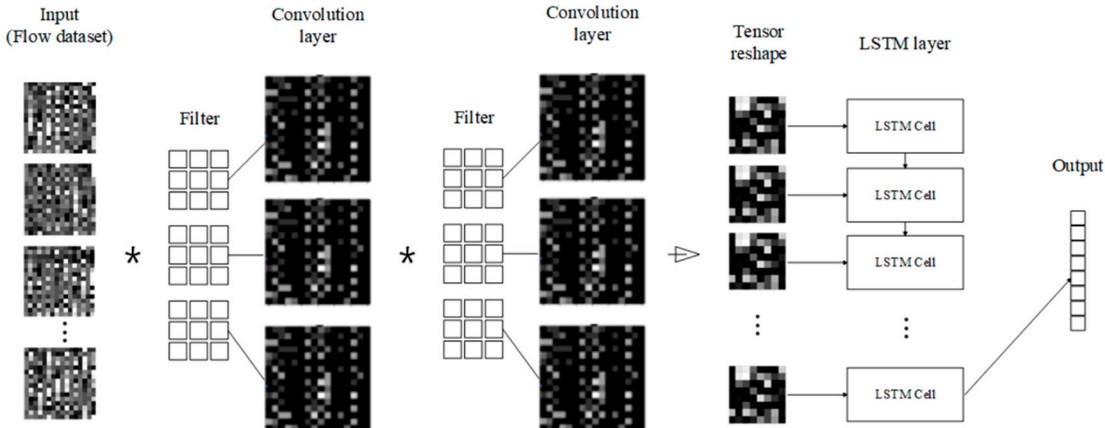

**Figure 4.** The convolutional neural network (CNN) and LSTM combination learning model architecture. It consists of two convolution layers, reshaping process, and single-layer LSTM.

## 6. Model Tuning

Each of the models has various hyper-parameters that determine the network structure (e.g., number of filters) and how the network models are trained (e.g., type of optimizer). The performance of a model can vary considerably according to the selected set of hyper-parameters. Toward this end, we use the grid-search [36] as the method to find hyper-parameters optimized for each deep learning model, according to the datasets. The grid search method searches for the best hyper-parameter for a dataset by trying every possible combination of hyper-parameters based on the dataset. We also verify the validity of the model by performing k-fold cross-validation [37] in addition to finding the optimal hyper-parameters. In k-fold cross-validation, the dataset is randomly partitioned into k equal-sized sub-datasets. Of the k sub-datasets, a single sub-dataset is retained as the validation data for testing the model, and the remaining k-1 sub-datasets are used as training data. The cross-validation process is then repeated k times, with each of the k sub-datasets used exactly once as the validation data. The k results can then be averaged to produce a single estimation. The advantage of this method over repeated random sub-datasets is that all observations are used for both training and validation, and each observation is used for validation exactly one time.

Figure 5 shows the overall process of evaluating a selected deep learning model with optimal hyper-parameters. First, we separate the flow-based dataset into learning and test data. Next, the learning data is separated into training and validation data, and k-fold cross-validation based on a grid search is performed. It performs the verification of the model on the basis of the pre-set hyper-parameter set and the k-fold value. Then, the model is trained by using the optimal hyper-parameters and the model performance is measured using the test data.

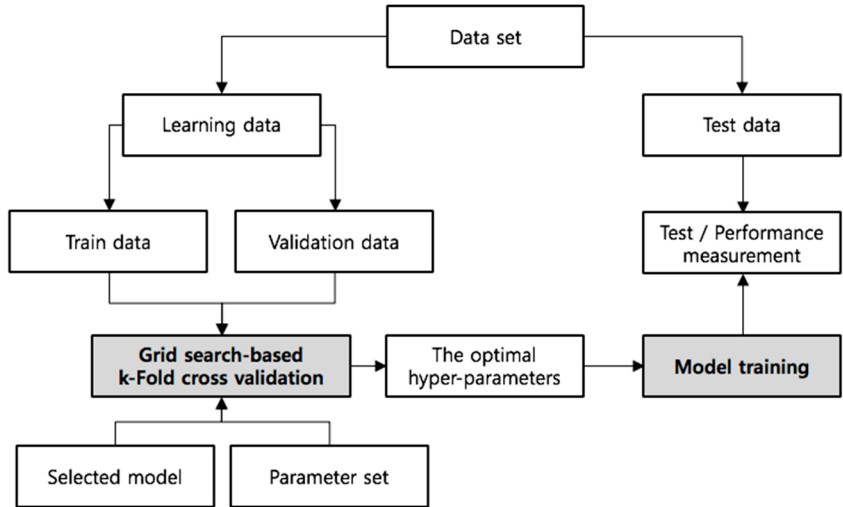

**Figure 5.** Process of hyper-parameter selection and model evaluation with the cross-validated grid-search.

*Multi-Layer LSTM and CNN + LSTM Model Tuning*

For the multi-layer LSTM and CNN + LSTM models, the flow-based datasets are constructed by arbitrarily fetching 2000 flows from the eight applications. In addition, we choose 36, 64, 256, and 1024 pixels as the imaged payload sizes of a packet in a flow. The number of sequential packets in a flow is set to 30, 60, and 100. Therefore, the final shapes of the datasets are as follows.

- (16,000, 30, 36), (16,000, 30, 64), (16,000, 30, 256), and (16,000, 30, 1024) for 30 packets in a flow
- (16,000, 60, 36), (16,000, 60, 64), (16,000, 60, 256), and (16,000, 60, 1024) for 60 packets in a flow
- (16,000, 100, 36), (16,000, 100, 64), (16,000, 100, 256), and (16,000, 100, 1024) for 100 packets in a flow

For the hyper-parameters of the multi-layer LSTM model, we consider (1) output size, (2) kernel initializer, (3) recurrent initializer, (4) dropout rate, (5) output activation type, (6) optimization type, and (7) batch size. The output size is the dimensionality of the output space. The kernel initializer represents the strategy to initialize the kernel weight vector values used for the linear transformation of the inputs. The recurrent initializer represents the strategy to initialize the weight vector values used for regularization. Furthermore, the dropout rate indicates the fraction of the hidden units to drop for the linear transformation of the recurrent state. The meanings of output activation type, optimization type, and batch size are the same as those used in any deep learning model. In the multi-layer LSTM models, the values used to perform the grid search for the hyper-parameters are listed in Table 1.

**Table 1.** The grid search hyper-parameter sets for the LSTM model.

| | Hyper-Parameter Values |
|---|---|
| output size | {64, 128, 256} |
| kernel initializer | {normal, uniform, glorot_uniform} |
| recurrent initializer | {normal, uniform, glorot_uniform} |
| dropout rate | {0.0, 0.2, 0.3, 0.4} |
| output activation type | {tanh, relu, softmax} |
| optimization type | {adam, rmsprop} |
| batch size | {1, 10, 100} |

For the CNN + LSTM model, a total of nine hyper-parameters are considered and listed in Table 2. The meaning of the seven hyper-parameters is the same as the ones in the LSTM model. The two hyper-parameters added for CNN + LSTM are as follows: (1) number of filters and (2) kernel size. The number of filters represents the number of the output filters of one convolution layer. The kernel size is the size of the kernel used in one filter.

**Table 2.** The grid search hyper-parameter sets for the CNN + LSTM model.

| | Hyper-Parameter Values |
|---|---|
| number of filters | payload size/2 |
| kernel size | $\{3 \times 3, 5 \times 5, 7 \times 7\}$ |
| kernel initializer | {normal, uniform, glorot_uniform} |
| output size | {64, 128, 256} |
| recurrent initializer | {normal, uniform, glorot_uniform} |
| dropout rate | {0.0, 0.2, 0.3, 0.4} |
| output activation type | {tanh, relu, softmax} |
| optimization type | {adam, rmsprop} |
| batch size | {1, 10, 100} |

For each size of the imaged payload (36, 64, 256, and 1024 pixels) and the selected number of sequential packets in a flow (30, 60, and 100 packets) in the multi-layer LSTM and CNN + LSTM models, Tables 3 and 4 show the optimal hyper-parameter values found through the cross-validated grid search process, respectively. As we can know from the tables, the optimal parameter values seem to be random for each hyper-parameter type. Nevertheless, we can find some pattern about the optimal parameter values. First, the output activation type is always "softmax" across all cases of payload size and the number of packets. The nonlinear logistic activation function can make the models' performance the best. Second, the "adam" optimizer produces the best model performance in most cases. Lastly, the model performance is enhanced when the batch size is relatively high (100 data samples).

**Table 3.** The optimal hyper-parameter values for the multi-layer LSTM model (U: uniform, N: normal, G: glorot uniform, S: softmax, R: rmsprop, A: adam).

| | Payload Size and Number of Packets per Flow | | | | | | | | | | | |
|---|---|---|---|---|---|---|---|---|---|---|---|---|
| | **36** | | | **64** | | | **256** | | | **1024** | | |
| | **30** | **60** | **100** | **30** | **60** | **100** | **30** | **60** | **100** | **30** | **60** | **100** |
| output size | 128 | 128 | 64 | 256 | 128 | 64 | 256 | 128 | 128 | 128 | 128 | 64 |
| kernel initializer | U | U | U | U | U | U | U | U | G | G | G | G |
| recurrent initializer | U | U | U | U | U | U | U | G | G | G | G | G |
| dropout rate | 0.2 | 0.2 | 0.2 | 0.2 | 0.2 | 0.2 | 0.2 | 0.2 | 0.4 | 0.2 | 0.3 | 0.4 |
| output activation type | S | S | S | S | S | S | S | S | S | S | S | S |
| optimization type | A | A | A | A | A | A | A | R | R | A | R | R |
| batch size | 100 | 100 | 100 | 100 | 100 | 100 | 100 | 100 | 100 | 100 | 100 | 100 |

**Table 4.** The optimal hyper-parameter values for the CNN + LSTM model (U: uniform, N: normal, G: glorot uniform, S: softmax, R: rmsprop, A: adam).

| | Payload Size and Number of Packets per Flow | | | | | | | | | | | |
|---|---|---|---|---|---|---|---|---|---|---|---|---|
| | **36** | | | **64** | | | **256** | | | **1024** | | |
| | **30** | **60** | **100** | **30** | **60** | **100** | **30** | **60** | **100** | **30** | **60** | **100** |
| number of filters | 18 | 18 | 18 | 32 | 32 | 32 | 128 | 128 | 128 | 512 | 512 | 512 |
| kernel size | $3 \times 3$ | $3 \times 3$ | $3 \times 3$ | $3 \times 3$ | $3 \times 3$ | $5 \times 5$ | $5 \times 5$ | $5 \times 5$ | $5 \times 5$ | $5 \times 5$ | $7 \times 7$ | $7 \times 7$ |
| output size | 64 | 64 | 64 | 256 | 128 | 64 | 256 | 128 | 128 | 128 | 128 | 64 |
| kernel initializer | G | G | G | G | G | U | U | U | G | G | G | G |
| recurrent initializer | N | N | U | U | U | U | U | G | G | G | G | G |
| dropout rate | 0.0 | 0.0 | 0.0 | 0.0 | 0.0 | 0.0 | 0.0 | 0.2 | 0.2 | 0.2 | 0.3 | 0.3 |
| output activation type | S | S | S | S | S | S | S | S | S | S | S | S |
| optimization type | A | A | A | A | A | A | A | A | A | A | A | A |
| batch size | 100 | 100 | 100 | 100 | 100 | 100 | 100 | 100 | 100 | 100 | 100 | 100 |

## 7. Experimental Evaluation

In this section, we compare the model prediction performance across the two models.

### 7.1. Experimental Environment

Our experiments are executed on Ubuntu 16.04 LTS with 32 GB of RAM and two GPU cards (NVIDIA GTX 1080Ti 11 GB). For the experimental implementation, we used Tensorflow-gpu 1.8 and Keras 2.2.0 operated with Python 3.6. The two models are constructed, trained, and tested by Keras using the Tensorflow-gpu backend.

According to the process shown in Figure 5, the three-fold cross-validated grid search is first performed for one dataset (k = 3) and the optimal hyper-parameters are found through each model tuning. The model training is performed using the found optimal hyper-parameters for each dataset, as listed in Tables 3 and 4. For such model training, we use the hyperbolic tangent function as the activation function of the LSTM layer. The number of learning epochs is set to 200. Lastly, the learning rate of optimizers (i.e., rmsprop and adam) is set to 0.001.

### 7.2. Performance Metrics

An unambiguous and thorough way to present the prediction results of a deep learning mode is to use a confusion matrix. Table 5 shows the confusion matrix produced by the test process of the multi-layer LSTM model with the optimal hyper-parameters and a flow-based dataset. For example, the model correctly predicts 681 cases of RDP, 571 cases of Skype, 670 cases of SSH, 694 cases of BitTorrent, 642 cases of HTTP-Facebook, 637 cases of HTTP-Wikipedia, 656 cases of HTTP-Google, and 598 cases of HTTP-Yahoo. It also misclassified 250 cases (all cases outside the diagonal positions in Table 5) out of the total number of all predicted cases. It also shows that the F1-scores of each application label retain at the high level with strong robustness, which confirms that the multi-layer LSTM model can effectively and stably distinguish network application.

**Table 5.** The multi-class confusion matrix by the multi-layer LSTM model test with the dataset of 60 sequential packets per flow and 1024 pixels of payload size (The bold line is used to convert this multi-class confusion matrix into the binary confusion matrix of the RDP application label shown in Table 6).

| Application Label | | Predicted | | | | | | | | Sum | F1-score |
|---|---|---|---|---|---|---|---|---|---|---|---|
| | | RDP | Skype | SSH | BitTorrent | HTTP-Facebook | HTTP-Wikipedia | HTTP-Google | HTTP-Yahoo | | |
| | RDP | 681 | 1 | 0 | 0 | 0 | 0 | 0 | 0 | 682 | 1.00 |
| | Skype | 0 | 571 | 0 | 11 | 0 | 0 | 0 | 0 | 582 | 1.00 |
| | SSH | 2 | 0 | 670 | 0 | 0 | 0 | 0 | 0 | 672 | 1.00 |
| **Actual** | BitTorrent | 0 | 2 | 0 | 694 | 0 | 1 | 0 | 0 | 697 | 1.00 |
| | HTTP-Facebook | 0 | 1 | 3 | 0 | 642 | 25 | 1 | 2 | 674 | 0.97 |
| | HTTP-Wikipedia | 0 | 0 | 0 | 0 | 45 | 637 | 1 | 26 | 709 | 0.97 |
| | HTTP-Google | 0 | 0 | 0 | 0 | 26 | 6 | 656 | 0 | 688 | 0.99 |
| | HTTP-Yahoo | 0 | 0 | 0 | 0 | 40 | 57 | 0 | 598 | 695 | 0.98 |
| | | | | | | Overall F1-score | | | | | 0.98 |

**Table 6.** The binary confusion matrix for the RDP application label in the multi-layer LSTM model test.

| | | Predictive | |
|---|---|---|---|
| | **n = 5400** | **Positive** | **Negative** |
| **Actual** | Positive | 681 (True Positive: TP) | 1 (False Negative: FN) |
| | Negative | 2 (False Positive: FP) | 4716 (True Negative: TN) |

However, the accuracy can be misleading when there is an imbalance on the number of application labels. A model can predict the label of the majority application for all predictions and achieve a high classification accuracy, and the model is not useful in the problem domain. For every application label, therefore, we convert the multi-class confusion matrix into the binary confusion matrix. Table 6 is an example of the binary confusion matrix for the RDP application label, which is derived from Table 5.

In the binary confusion matrix described by Table 6, we observe that, out of the total of 5400 RDP prediction cases, the multi-layer LSTM model predicts 683 (= 681 + 2) cases as RDP, and predict 4717 (= 1 + 4716) cases at the others. In actual, 682 (= 681 + 1) test datasets are RDP, while 4718 (= 2 + 4716) test datasets are the others. The true positive (TP) indicates the cases in which the actual label is positive (RDP) and the model prediction is also positive correctly. The false negative (FN) indicates the cases in which the actual label is positive, but the model prediction is negative incorrectly. The false positive (FP) indicates the cases in which the actual label is negative (that is, not RDP), but the model prediction is positive incorrectly. Lastly, the true negative (TN) indicates the cases in which the actual label is negative and the model prediction is also negative correctly.

To overcome the problem of accuracy measurement, we compute the F1-score [38] as well as accuracy to evaluate the two models. They are defined by using the binary confusion matrix as follows.

$$Accuracy = \frac{TP + TN}{TP + TN + FP + FN} \tag{1}$$

$$F1-score = \frac{2 \times Precision \times Recall}{Precision + Recall} \tag{2}$$

where *Recall* = *TP*/(*TP* + *FP*) and *Precision* = *TP*/(*TP* + *FN*). The F1-score represents the harmonic mean of precision and recall, and indicates the classification performance of a deep learning model relatively accurately. We compute the accuracy, recall, precision, and F1-score values of all eight application labels. Then, we average them to get the overall accuracy and F1-score measurements for performance comparison of the two models.

*7.3. Experimental Results*

The experiment is performed to compare the performance of the multi-layer LSTM and CNN + LSTM models with the flow-based datasets. Figure 6 shows the comparison of the overall accuracy and F1-score values for the two models with the eight application labels in terms of the three numbers of packets per flow and the four imaged payload sizes. It can be seen in the figure that the overall accuracy and F1-score values increase with the payload size of packets. Furthermore, as the number of sequential packets per flow increases, they increase, too. As shown in Figure 6a, the overall accuracy of the multi-layer LSTM model is higher than that of the CNN + LSTM model. When the number of packets per flow is 30 and the payload size is 36, the accuracy values of the multi-layer LSTM and the CNN + LSTM are 61.66% and 60.96%, respectively. On the other hand, when the number of packets per flow is 100 and the payload size is 1024, the corresponding accuracy values are 99.65% and 98.86%, respectively. With F1-score values, similar results are shown in Figure 6b. When the number of packets per flow is 30 and the payload size is 36, the F1-score values of the multi-layer LSTM and CNN + LSTM models are 0.89375 and 0.885, respectively, which are the lowest values. On the other hand, when the number of packets per flow is 100 and the payload size is 1024, the corresponding F1-score values are 0.99575 and 0.9925, respectively, which are the highest values.

We also know that the F1-score of the multi-layer LSTM model is higher than the one of the CNN + LSTM model for all cases of the number of packets per flow and the payload size of packets. As the payload size increases, the F1-score values of the multi-layer LSTM and CNN + LSTM models become similar. For most cases of payload size, however, the multi-layer LSTM's F1-score is higher and more stable than the CNN + LSTM's one when many sequential packets in a flow are used for the datasets. That is, we know that the multi-layer LSTM model alone can provide good performance.

Figure 7 shows the comparison of the overall F1-score values for the number of LSTM layers in terms of two datasets: (1) the smallest dataset of 30 packets per flow and 36 payload sizes (Figure 7a), and (2) the largest dataset of 100 packets per flow and 1024 payload sizes (Figure 7b). As shown in Figure 7a, the F1-score values of the single-layer, two-layer, and three-layer LSTM models are 0.875, 0.89375, and 0.9875, respectively. The multiple LSTM layers help increase the model performance. The similar results are shown in Figure 7b and we can also know that the two-layer or three-layer

LSTM models can classify the network traffic almost completely when the training data set is large enough. The corresponding F1-score values are 0.99575 and 0.9975, respectively, while it is 0.9625 when a single-layer LSTM model is used.

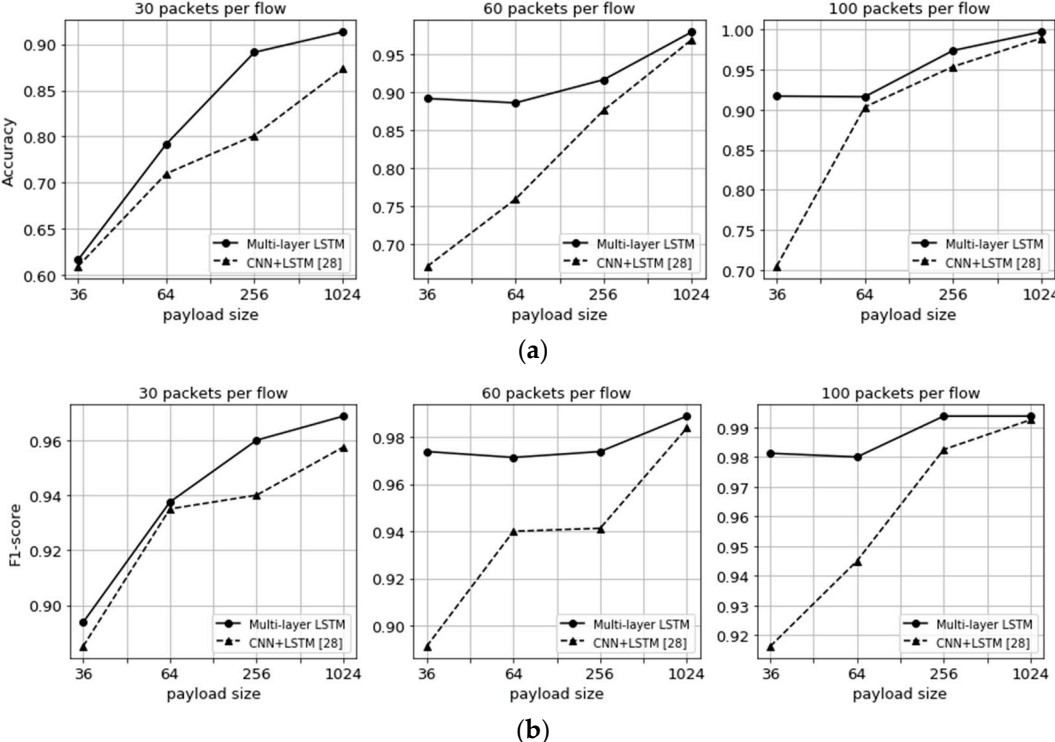

(a)

(b)

**Figure 6.** Performance comparison of the multi-layer (two-layer) LSTM and CNN + LSTM [28] models with accuracy and F1-score values in terms of the three numbers (30, 60, and 100) of packets per flow and the four imaged payload sizes. (**a**) Accuracy and (**b**) F1-Score.

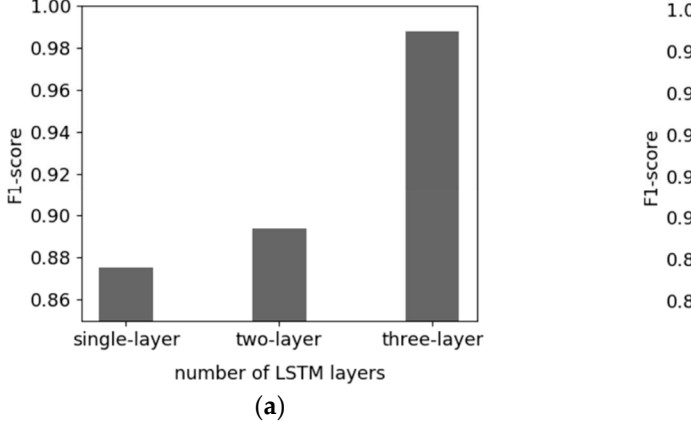
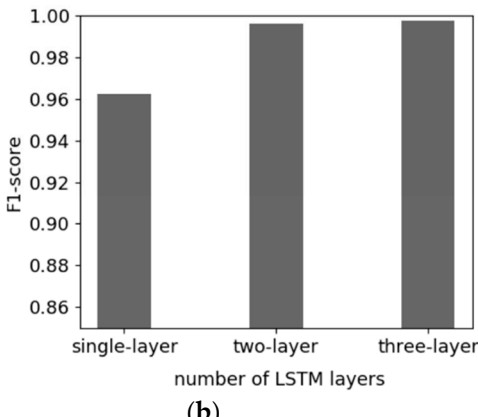

(**a**)    (**b**)

**Figure 7.** Performance comparison of the single or multi-layer (two or three-layer) LSTM models. The figure on the left shows the F1-score comparison based on the number of LSTM layers using the dataset of 30 per packet flows and 36 payload sizes. The figure on the right shows it using the dataset of 100 per packet flows and 1024 payload sizes. (**a**) The smallest dataset, and (**b**) the largest dataset.

In the CNN + LSTM model, the convolution layers extract the features from the payload data, and then the features are used for the input data to the LSTM layer. Although the features extracted from the convolution layers contain important information for the traffic classification, we know that the unchanged original payload itself can be useful for the traffic classification of the unchanged information located at the front of a flow is well utilized to classify the flow. The multi-layer LSTM

model is a special kind of recurrent neural networks capable of learning the long-term dependency over the sequence of packets in the front of a flow. In addition to that, the multi-layer LSTM model has multiple hidden LSTM layers where each layer contains multiple memory cells. In general, the multi-layer LSTM model learns the low-level features of traffic data in the initial layer and then the high-level representation of traffic data in the upper layer. The deeper multi-layer LSTM is used, the better it performs usually [39,40].

## 8. Discussion: Flow Classifier in SDN

With the learned multi-layer LSTM model, it is necessary to suggest ways to achieve the essential purpose of this paper. In this section, therefore, we describe how to classify traffic flow using the learned model in the flow classifier of the proposed SDN-based network architecture (see Figure 1).

The OpenFlow protocol should be extended to feed the payload information to the traffic classification model. When a packet arrives at an Open vSwitch, the switch looks up its flow table and executes an action such as dropping or forwarding to the packet. If there is no action rule in the flow table, the packet-in message is sent to the controller. The packet-in message is expanded such that it contains the beginning of the packet's payload. That is, the pre-defined 36, 64, 256, or 1024 pixels from the front of the packet payload are contained in the packet-in message (one pixel indicates the four bits of the payload data). To use our LSTM model, consecutive $N$ packet payloads per flow are needed ($N = 30$, 60, or 100). Therefore, the controller continually sends to the Open vSwitch the packet-out message containing 'null action' rule until receiving $N$ packet-in messages for the packet arrivals in the same flow. The 'null action' indicates that the Open vSwitch does not create any action rule for the packet. When the controller receives the $N$-th packet-in message including packet information generated in the same flow, the sequence of $N$ packet's payload information is fed into the learned multi-layer LSTM model and the flow classification is performed. At this point, the controller sends to the Open vSwitch the packet-out message containing rules, according to the scheduling or QoS policy based on the classified flow.

## 9. Conclusions

In this paper, we first propose an SDN-based network architecture for traffic classification using deep learning. In the SDN control plane, a well-trained traffic classifier can detect traffic of data plane without a network administrator and helps the QoS module apply new flow rules into OpenFlow switches. In such an SDN architecture, we propose traffic classification schemes using the two deep learning models: multi-layer LSTM and CNN + LSTM. Imaged packet payload data are generated through our own pre-processing method, and the flow-based payload dataset is created by gathering such imaged packets for each of the eight applications. We also execute the cross-validated grid search to find the optimal hyper-parameters that maximize the performance of the deep learning models. Through our intensive experiments, we know that the deep learning models can classify the network traffic fairly well, and the multi-layer LSTM model performs better than the CNN + LSTM model. Our dataset consists solely of the payload of the packets. In the CNN + LSTM model, the features extracted from the convolution layer are used for the input data to the LSTM layer. However, the unchanged original payload itself is more useful for the traffic classification if the problem of the long-term dependency is handled by a proper strategy. We can conclude that the multi-layer LSTM model can solve the problem in the case of the network traffic classification.

**Author Contributions:** Conceptualization, H.-K.L., Y.-G.H., and Y.-H.H. Data curation, J.-B.K. Formal analysis, J.-B.K. and Y.-H.H. Investigation, K.K. Methodology, J.-B.K. and Y.-G.H. Resources, K.K. Software, H.-K.L. Supervision, Y.-G.H. Validation, H.-K.L., J.-B.K., and K.K. Writing – original draft, H.-K.L. and Y.-H.H. Writing – review & editing, Y.-H.H.

**Funding:** Two Basic Science Research Programs through the National Research Foundation of Korea (NRF) funded by the Ministry of Education (2018R1A6A1A03025526 and 2016R1D1A3B03933355) supported this research.

**Conflicts of Interest:** The authors declare no conflicts of interest.

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
