# Peer review of "Payload-Based Traffic Classification Using Multi-Layer LSTM in Software Defined Networks†"

_applsci, doi:10.3390/app9122550_

Round 1
Reviewer 1 Report
This paper proposes a deep learning approach for the classification of network traffic, based only on payload data. The proposed idea consists in integrating a software defined network with two customized blocks performing data labeling and classification. Authors tested two powerful architectures: a multilayer recurrent network based on LSTM units and a CNN+LSTM network. Numerical results, show that the multilayer LSTM network provides very good results in terms of F1-score
The paper is an extended version of a conference paper. The paper is interesting and well organized. However, some issue should be addressed.
1) In the whole paper: “f1-score” => “F1-score”
2) Authors should underline the differences and novelty with respect to their previous work in [37].
3) Authors should better underline the differences and novelty with respect to the work in [27].
4) Figure 2 is missing from the paper.
5) Page 9, line 313: “non-linear” => “nonlinear”
6) In Tables 3 and 4, is the “softmax” activation type maybe meaning for the hyperbolic tangent activation function? If so, please revise.
7) Materials at page 13, lines 352-366 is well known and can be reduced.
8) Results in terms of only F1-score could be limited. Other metrics could be used jointly (accuracy, precision, recall, AUC, etc.).
9) It could be useful to provide some numerical results on the CNN+LSTM approach by varying the number of convolution layers.
10) It could be also helpful to provide results oa s CNN network alone, without the last LSTM layer.
11) In reference [17] the 5-th authors should begin with a capital character.
Author Response
First of all, we would like to thank you for all that you commented for our paper.
I attach the response to the comment from the reviewer as a PDF file.

Reviewer 2 Report
Contribution: This paper first proposes an SDN-based network architecture for traffic classification using deep learning. Imaged packet payload data are generated through our pre-processing method, and the dataset is created by gathering such imaged packets for eight tested applications. In such an SDN architecture, the authors introduce traffic classification schemes to use two deep learning models: multi-layer LSTM and CNN+LSTM. They also find the optimal hyper-parameters to maximize the performance of the deep learning models. After experiments, they find deep learning models can classify the network traffic fairly well. They also find multi-layer LSTM model performs better than the CNN+LSTM model in the experiments.
Comments and Suggestions:
In this paper, the authors did not tell us why the unchanged original payload itself is more useful for the traffic classification than the features extracted from the convolution layers of CNN+LSTM. Please give more explanation about this issue.
Some related works are missing. For traffic scheduling and QoS control in SDNs, I think the following papers “Traffic scheduling for deep packet inspection in software‐defined networks” and “Joint Optimization of Rule Placement and Traffic Engineering for QoS Provisioning in Software Defined Network” should be discussed in the section of related work.
In the section of Experimental Evaluation, I did not find anything about SDNs. At least one group of simulations should be conducted under the SDN environment, since the SDN is an important term in the title of this paper.
Author Response

(The authors gave the same response as above.)

Reviewer 3 Report
In the introduction, the authors claim that excluding header information in the training or learning model make the learned model fit unseen packets well and give the model a generalization ability. How to prove this claim? Do you have experiment result or analysis to show this statement?
The description of each work in the related work section are too high-level. It's better to provide each work's special features comparing to others and their quantitative result. Also, I only found two related works in this section are related to 'traffic classification using ML in SDN'. I think this topic is not quite new, there are many similar solutions and works already exist. It's better to compare the proposed work to others and emphasize contributions of the work.
The authors describe: 'The learned model can be used to classify network traffic in the flow classifier.' However, there is no paragraph showing the readers how the 'flow classifier' works!
In the SDN architecture, only the first packet will be sent to controller with the openflow protocol. In the proposed solution, it seems like you need to collect every packet in a flow and then send the bundled flow dataset to the classifier in the controller. Are these processes happen in both training and test phases? If they are, it will produce a serious negative impact to the network traffic throughput. Since the packets in the data plane will wait for the flow rule from controller. If not, please justify it.
In the table 5 and 6, what these values in the tables means and how to get these values? Please explain it.
Author Response

(The authors gave the same response as above.)

Round 2
Reviewer 1 Report
Authors have taken into account all my concerns and suggestion. The resulting revised manuscript is an enhanced version. I have no more comments.
Reviewer 3 Report
Thanks the authors for the response and revision. My doubts and questions of this article have been cleared.
This manuscript is a resubmission of an earlier submission. The following is a list of the peer review reports and author responses from that submission.
Round 1
Reviewer 1 Report
The article is at a good level and topic is interesting. However, many aspects should be improved. So I can recommend to accept paper after major revision.
In order to improve manuscript, I suggest the following recommendations:
R1: Please write clearly in the abstract about: 1) results (present and summarize the obtained result), and 2) conclusion.
R2: In section "Related Work" please add that deep learning methods (RNN, CNN etc.) can be effectively applied to problems from other fields, eg articles:
- "Approximation of Phenol Concentration using Computational Intelligence Methods Based on Signals from the Metal Oxide Sensor Array",
- "Classification of tea specimens using novel hybrid artificial intelligence methods",
- "Arrhythmia detection using deep convolutional neural network with long duration ECG signals",
- "Novel Deep Genetic Ensemble of Classifiers for Arrhythmia Detection Using ECG Signals",
- "Approximation of phenol concentration using novel hybrid computational intelligence methods".
R3: In the end of introduction section, please write clearly: 1) list the innovative elements and 2) identify motivations for undertaking research.
R4: Please add a table with detailed information about the structure and parameters of proposed methods (LSTM and CNN).
R5: I suggest in future to use evolutionary computation (eg. genetic algorithm) to optimize parameters instead of grid search.
R6: Please also calculate f1-score for the confusion matrix from table 5.
R7: Please add deeper conclusions based on the results obtained.
R8: Why was k = 3 used, not standard k = 10 in cross validation?
R9: Please, write how much time for the proposed methods (LSTM, CNN), optimization, training and classification (of single sample) stage lasted.
R10: Please compare the proposed method with the methods of other authors (state of art comparison table).
R11: Please add the results for some classical classification method, e.g. SVM, and then compare the results for: SVM, LSTM and CNN + LSTM in the table.
R12: Please in the conclusion section: 1) list the advantages and disadvantages of the proposed solution, 2) indicate the limitations of work, and 3) describe future works.
Reviewer 2 Report
This paper presents a methodology for constructing the SDN-based network architecture. To do this, authors adopt two new sub-systems, which are ‘labeled packet collector’ and ‘classifier’ in the data and control planes, respectively. To show the efficiency of the proposed scheme, authors employ deep learning-based test methods. Even though they provide various experimental results based on their own network system, there are severe technical issues as follows :
1) First of all, the main contribution of this paper is not clear. Authors emphasize the deep learning-based traffic classification (even in the title of this paper), however, the new point corresponds to the network architecture as shown in Fig. 1. Two deep learning-based schemes are just employed for evaluating their SDN-based network architecture. Therefore, authors should focus on emphasizing their new architecture rather than “deep learning”-based test schemes. What is the main difference between their architecture and other SDN-based networks ? To be clear, authors need to rewrite the beginning parts of the proposed method by highlighting their contributions.
2) Regarding experimental results, authors show the performance variations according to parameter setting, however, there is no performance comparison with other approaches. This makes the proposed method less convincing. Authors need to add the comparative studies both qualitatively and quantitatively.